# Evaluating the Injection Moulding of Plastic Parts Using In Situ Time-Resolved Small-Angle X-ray Scattering Techniques

**DOI:** 10.3390/polym14214745

**Published:** 2022-11-05

**Authors:** André A. Costa, Fábio Gameiro, Artur Potêncio, Daniel P. da Silva, Pedro Carreira, Juan Carlos Martinez, Paula Pascoal-Faria, Artur Mateus, Geoffrey R. Mitchell

**Affiliations:** 1Centre for Rapid and Sustainable Product Development, Polytechnic of Leiria, 2430-080 Leiria, Portugal; 2NCD-SWEET Beamline, Alba Synchrotron Light Source, Cerdanyola del Vallès, 08290 Barcelona, Spain

**Keywords:** injection moulding, crystallisation, flow, small-angle X-ray scattering, isotactic polypropylene

## Abstract

In this study, we describe the design and fabrication of an industrial injection moulding system that can be mounted and used on the NCD-SWEET small-angle X-ray scattering beamline at ALBA. We show how highly useful time-resolved data can be obtained using this system. We are able to evaluate the fraction of the material in the mould cavity and identify the first material to solidify and how this varies with the injection temperature. The design follows current industrial practice and provides opportunities to collect time-resolved data at several points within the mould cavity so that we can build up a 4D perspective of the morphology and its temporal development. The quantitative data obtained will prove invaluable for the optimisation of the next generation of injection moulding techniques. This preliminary work used results from the injection moulding of a general-purpose isotactic polypropylene.

## 1. Introduction

Injection moulding is the most common technology employed to fabricate parts from polymers [1]. In this, a polymer in the softened or molten state is injected, at high pressure, into a shaped mould, usually manufactured in metal and held at room temperature or an intermediate temperature. As a consequence, the polymer cools and becomes solid either through a glass transition or a crystallization process. Semi-crystalline polymers such as polyethylene and polypropylene are the most widely used, and the time profile of the temperature and flow have a major impact on the properties of the final part [2]. The presence of a flow inherent in the injection moulding process, means that the final product does not show the typical spherulitic morphology of crystallisation from a quiescent melt [3]. In the case of the crystallisation of a melt subject to flow [4], the longest chains of the polymer melt become extended, and these serve as row nuclei for the unstressed chains in the polymer melt, which leads to a high level of preferred orientation in the lamellar crystals due to the common alignment of the extended chains. Of course, the generation of a highly aligned morphology depends on whether the extended chains have relaxed back to an isotropic configuration, and this, in turn, will depend on the molecular weight distribution of the polymer. In the standard tube model of rheology, relaxation time is needed for a chain constrained in a tube scale with the molecular length squared. All of these factors mean that the morphology which develops during injection moulding involves an inter-play of the cooling rates, flow conditions, and relaxation. For those reasons, the development of morphology has a history, and some post-moulding experimental techniques, such as the electron microscopy of differentially etched samples [5,6], in skilled hands can unravel the history.

Traditionally, crystallization and flow are studied with laboratory equipment and the post-moulding analysis of samples prepared using injection moulding. In the last 30 years, synchrotron-based small-angle (SAXS) and wide-angle X-ray scattering (WAXS) [7] have greatly enhanced our understanding of polymer crystallization using in situ time resolving techniques and where appropriate flow stages, such as shear cells [8,9]. SAXS techniques have the capability to quantify the nature and the development of crystals and the level of preferred orientation [6]. Now, as outstanding as this work was, it had limited direct relevance to industrial injection moulding practice due to the extreme cooling rates and the interwoven flow fields present in the mould. We reviewed all of the possible time-resolving quantitative techniques and identified that synchrotron-based SAXS techniques provided the only practical technique which could be used. They provide quantitative data on the relevant structural scales, the energy of the X-rays means for which a relevant sample thickness could be used, do not require optical transparency, the small diameter of the incident beam means that data are obtained which are relevant to a specific point meaning that mapping an area would be straightforward, and the small volume of the incident and scattering beams results in a simple design strategy. When we reviewed the literature, we found very few relevant publications. There were, of course, various igneous experiments to study very fast processes using calorimetry in conjunction with SAXS measurements, such as [10] which provides an excellent example using microbeam SAXS to map out the morphology of injection moulded parts [11] and the combination of SAXS and extensional rheology [12]. However, we were only able to find a few publications relevant to in situ measurements and injection moulding. Some had employed WAXS techniques [13,14], which have different requirements to SAXS to study the moulding of liquid crystal polymers. The most relevant work was on micro-injection moulding employing metal moulds and X-ray transparent diamond windows [15,16]. Diamond windows are widely used in high-pressure cells for X-ray scattering [17], but of course, they give access to only a small part of the mould cavity. We identified these publications as very interesting works, but which were focused on developing a scientific understanding rather than reflecting industrial practice.

This work is focused on addressing this matter in order to design, manufacture and test an injection moulding system, which is designed to be relevant to current and, as far as possible future industrial practice.

## 2. Materials and Methods

The design of an injection moulding system to mount on the selected beamline [18] must meet the geometric and weight limitations of the beamline and its components. After consideration of the various options, we selected an Industrial Autonomous Injection Unit, UAI6/1OP, from the Rambaldi Group, which consolidates most of the control and power components and the remote control in a floor-mounted unit. The mould and clamping systems were designed by us. The mould cavity was manufactured from standard mould components, mould plates, and mould inserts and used industrial-scale recirculating heating and cooling units to maintain a constant temperature. The key components of the mould design and its relationship to the incident X-ray beam are shown in Figure 1.

This work selected and utilized the NCD-SWEET SAXS Beamline at the ALBA Synchrotron Light Source, which is equipped with a DECTRIS Pilatus3S 1 M detector system mounted at a variable distance from the sample position. In this work, the sample to the detector distance was set at 6.7 m, and we used an incident wavelength of 1 Å with the equivalent photo energy of 12.4 KeV. The Pilatus Detector system was built up of an array of silicon sensors equipped with CMOS electronics. As a consequence, a small portion of the detector was not active (~7%), which appeared as black stripes in the recorded intensity images. The charge induced by the X-ray photons was detected and processed by the pixel readout system. This detector had an effective pixel size of 172 × 172 micrometres and a dynamic range of 20 bits. SAXS patterns were recorded over a |Q| range from 0.002 Å^−1^ to 0.125 Å^−1^ where |Q| = 4π Sinθ/λ, 2θ is the scattering angle, and λ is the incident wavelength. Although we employed a fixed geometry with a flat detector, the small scattering angle meant that the scattering vector effectively lay in the plane of the mould cavity, and so the complete azimuthal range of data, α = 0 to 360° were available to a static detector, where α is the angle between the scattering vector Q and the flow axis in the mould cavity as shown in Figure 2. To prevent saturation of the detector by the zero-angle X-ray beam, a beam stop was placed in front of the detector to absorb the transmitted beam. The detector orientation and sample-to-detector distance were calibrated using the well-known standard silver behenate [19].

This work focused on the moulding of the isotactic polypropylene, specifically Lyondell Basell Polypropylene MOPLEN HP500N, which is a general-purpose polypropylene suitable for food contact applications. It exhibits a melt flow index (MFI) of 12 g/10 min at 230 °C. The injection moulding system has a minimum charge of pellets for a successful operation of ~12 g, which makes the system particularly suited for the research and development of new materials. The X-ray computed tomography (CT) scans of the mould inserts were obtained using a Nikon XT H 225 ST 2X Industrial Scanner (Nikon Metrology Europe NV, Leuven, Belgium).

The X-ray beam paths of both the incident beam and the scattered signal, were a critical part of the injection moulding system. There are competing factors of X-ray transparency and thermal conductivity together with mechanical strength to withstand the pressure in the mould, which may reach 1600 Bar. There have been relatively few systems reported in the literature, and one approach chose to reconcile these competing requirements by opting for a metallic mould with X-ray transparent windows in the form of diamonds [15,16]. This is a well-established approach widely exploited in the design of X-ray cells for high-pressure studies [17]. We noted that one of the key differences between a high-pressure stage and an injection moulding cavity was the scale of the active volume and the need to probe the behaviour of the polymer throughout the cavity.

We opted to follow an alternative approach that maintained the current mould design practice while introducing a high level of X-ray transparency. This is a development of the approach employed by Rendon et al. [14]. Figure 1 shows the two mould inserts employed to make the mould cavity and the holes machined to provide the “windows”. Figure 3 shows the CT scan of one of the mould inserts used to define the cavity.

These inserts are made of a standard mould insert material, an aluminium/silicon alloy AW6082. Six blind holes were introduced into each of the two mould inserts. Initially, the hole had a diameter of 5 mm, and, close to the mould face, the hole was narrowed to 1.5 mm to minimize the area of the “window” and the hole was stopped at ~0.1 mm from the mould surface. The CT image confirmed the expected results, and the thickness of the 12 “windows” were measured as 0.0832 mm with a standard deviation of 0.0015 mm. This gave an attenuation for the two inserts of <|30|% for 12.4 keV photons. Changes to the cavity geometry or the thermal characteristics can be quickly introduced by replacing the inserts. They are located in a precision-engineered slot and retained using screws. The hardening of the aluminium alloy from which the inserts were made was studied by Rowolt et al. [20]. The temperature cycle that the inserts experience during the injection moulding experiments does not affect the level of precipitation in the alloy, which could lead to a change in this “background scattering”. The choice of the photon energy is a compromise between efficiency in the photon production, photon detection, attenuation in the windows (and air), and the desired |Q| range. The attenuation of the incident X-ray beam by the “windows” is dependent on the photon energy. This energy is a characteristic of the selected beamline. 

In this work, we have chosen the NCD-SWEET SAXS-WAXS beamline at ALBA. The source for NCD_SWEET is an in-vacuum undulator, the maximum flux is available at 12.4 keV, and although the source can be tuned to specific wavelengths in the range 0.62 to 1.90 Å, the wavelengths shorter than 1Å have a lower flux for this particular undulator. Now, higher energy or shorter wavelength beams will exhibit a lower attenuation with the aluminium alloy of the mould inserts than for 12.4 keV, but this is not the only factor. The incident beam flux is a critical factor, and the final signal produced will also depend on the sensitivity of the detector. The Pilatus-type detector is sensitive over a range of photon energies, but for high energies, it requires a thicker silicon slice at the front of the detector to ensure the full absorption of the photons. Using a higher photon energy also modifies the scattering angle to about 50% of the value with 12.4 keV photons. As we are currently using the maximum sample to detect distance, this compresses the SAXS into a smaller region of the detector. In short, the beamline is optimised for 12.4 keV photons, and as the attenuation is quite reasonable, we selected to use that energy. Other facilities, such as PETRA in Germany, provide SAXS facilities on beamline P62 with higher photon energies from 3.5 to 35 keV [21], and the Diamond Light Source in the UK provides SAXS on beamline I12 JEEP with higher photon energies of 53–150 keV [22].

## 3. Results

The injection moulding system was successfully mounted on the NCD-SWEET Beamline (Figure 4). The system was aligned with respect to the incident X-ray beam using the xyz translation stage in conjunction with a diode which measured the incident X-ray intensity in front of the beam stop. The injection moulding system was also mounted on a rotational stage, and this proved invaluable for aligning the relatively long and narrow pathway through the mould. After alignment, we then measured the scattering from the empty cavity. It exhibited a well-defined isotropic pattern centred around the beam stop (Figure 5a), which is typical for an aluminium/silicon alloy that forms the “window” material. Figure 5b shows the azimuthally averaged data. This scattering can be easily subtracted from the experimental patterns when the mould contains the polymer. It is clear that the alignment of the system was very good, and there were no specular reflections from the incident beam hitting the side of the blind holes in the mould inserts. The alignment of the systems was stable when the mould was opened and closed and also when the polymer was injected under pressure into the mould.

All of the equipment worked according to plan, including the remote control of the injection moulding system and the ejection of the part. We were able to observe remotely in real-time the temperature and pressure in the mould. A part could be produced on a cyclic basis using the remote control. Each of the six windows allowed the development of the morphology to be evaluated at different points in the mould cavity.

We were able to identify three types of experiments (Table 1). The first type, Mode A, involved positioning the beam at one of the windows, for example, window one, and recording a time-resolving sequence of small-angle X-ray scattering patterns from before the start of the injection process to the completion of the moulding cycle. The second type of experiment also used Mode A for a similar sequence involving a succession of equivalent moulding cycles, but with the incident X-ray beam positioned in turn at each of the windows. In such an arrangement, we relied on the reproducibility of the injection moulding cycle so that we could sample different points in the cavity in different moulding cycles. Alternatively, we could relatively move the incident beam during one cycle to different windows, but the movement time would reduce the amount of scattering data recorded for each cycle and provide an interval of 18s between successive measurements at the same window after cycling through the other five windows. One version of this was Mode B, but we could tailor the sampling points and sequences optimised to yield specific data (Mode C). During each data accumulation, we measured the transmission of the cavity using a diode placed in the beam-stop, and this enabled the fraction of the material, whether molten or solid, in the mould cavity to be evaluated. In the experiments reported here, we focused on a time series of patterns recorded for a particular window (Mode A); multiple window experiments will be reported in a later publication after the required experiments. We note here that the position of the sampling ports can be adjusted for a particular mould cavity or processing parameters by preparing a set of mould inserts with different port positions guided, for example, by computer simulations.

In Figure 6, we show SAXS patterns obtained using Mode A with a sampling position of one, which is the closest position to the injection point. Figure 6a shows the sequence of SAXS patterns recorded 10 s after the injection of the polypropylene at 250 °C. The mould temperature was held constant throughout these experiments at 50 °C. The first pattern in Figure 6a is that of an empty cavity. As we move to the right in this sequence, we can observe the emergence of a circular ring of scattering. The pattern is typical for a semi-crystalline polymer in which crystallisation has been nucleated in a quiescent melt, and the chain-folded lamellar crystals have grown in random orientations. The position of this intense peak can be used to evaluate the long period, and from the azimuthal variation in intensity, the level of the preferred orientation of the lamellar crystals can be identified, which, in this case, was zero. We used the azimuthal variation in intensity as a function of α and a fixed value of |Q| corresponding to the peak maximum.

The sequence of patterns shown in Figure 6d is very different from those in Figure 6a and shows the development of a highly anisotropic pattern in which the intensity is greatest in the direction parallel to the long axis of the mould cavity and to the direction of flow within that cavity. These patterns were recorded for an extrusion temperature of 190 °C and a constant mould temperature of 50 °C. The pattern is typical for a semi-crystalline polymer in which crystallisation has been nucleated in the melt by the row nuclei, and the chain-folded lamellar crystals have grown out normally to the row nuclei and, therefore, to the flow axis [4,23]. The very high level of orientation is a consequence of the alignment of the extended chains which form the row nuclei and which have a common axis of alignment throughout the sample in this central zone of the mould cavity. As we progress in time, we see the development of greater fractions of crystals, all of which have the same very high level of preferred orientation. We need to remember that the material will crystallise even under quiescent conditions, albeit at a slightly lower temperature. In this case, all of the crystals have been templated by the row nuclei, and the density of row nuclei is sufficiently high that the chain-folded lamellar crystals growing out from the row nuclei impinge and fill the space before the directing influence of the row nuclei is lost. The work of Bassett [24] shows elegantly that the directing influence can extend over 1µm in the case of polypropylene.

Now, let us consider the sequences obtained with intermediate injection temperatures. Sequence (c), was obtained with an injection temperature of 205 °C. Initially, the SAXS patterns showed the same behaviour as those obtained at 190 °C; however, in the fourth frame and beyond we could observe a component of the scattering that was less anisotropic than that in the first two frames. A broadly similar behaviour was observed for sequence (b) obtained with an injection temperature of 210 °C.

Figure 7 shows the sections taken through the 2D SAXS pattern recorded 100 s after the first observed crystallization for each of the experiments shown in Figure 6. The section was taken in the horizontal direction, passing through the centre of the detector and using the software package “Datasqueeze” [25]. The data were averaged over a small |Q| interval (5 × 10^−3^Å^−1^) in the vertical axis at each |Q| value. The sections taken through each SAXS pattern reveal a peak in the scattering, which corresponds to the scattering from the lamellar stacks with amorphous material between the lamellar crystals. The position of the maximum intensity, |Q_0_|, can be converted to the long period Lp using the following equation [26].
Lp = 2π/|Q_0_|(1)

We have applied this methodology to the first SAXS pattern in which there is a reliable signal to make measurements, and typically this was at 3 s. We also evaluated the long period for the patterns taken at 5 s and at 100 s (Figure 5), and these values are listed in Table 2. This gave a value of ~140 Å for the scattering recorded for an injection temperature of 190 °C. The SAXS patterns for other injection temperatures showed broadly similar values at t = 100 s, as can be observed in Table 2.

The measured long period for the first lamellar crystals varied in a reciprocal manner with the initial melt temperature. Now, as the injection temperatures were considerably higher than the quiescent crystallization temperature of the polymer [24], we might have expected that the major difference would have been that of the time to cool to the relevant crystallization temperature. In part, this effect is demonstrated in a simple manner in Figure 6. However, the different temperatures also impacted the flow behaviour and subsequent relaxation, which affected the number density of the extended chains in the cavity and defined the level of templating and, hence, the global orientation parameters. As time proceeded since the first crystals were formed, this inverse trend in the long period continued, although the values for the highest temperature lie outside of this trend. We should remember that crystallization within the mould is far removed from isothermal conditions, and so the morphology contains principal components that are developed at higher temperatures and infill lamellar, which are formed at lower temperatures. Attaching a single value to the long period for each measurement does not take into account any inhomogeneity through the thickness of the mould or in the volume measured. We anticipated that the morphological evaluation by differential etching and electron microscopy would provide invaluable information to guide the analysis, albeit post-moulding. After a prolonged cooling and crystallization time (100 s), the long-period values converged to broadly similar values. The most significant differences in the morphology probed by SAXS was the different levels of the preferred orientation of the lamellar chain folded crystals, as can be observed directly from the patterns shown in Figure 6.

We note here that the sequences of the SAXS patterns shown in Figure 6 are perhaps not the most useful display of the data, and that it would be more helpful if we could present the SAXS patterns which correspond to the crystallization which took place in each 1 s frame or in other words, we needed to unravel the accumulated data shown and separate it into the scattering for the first 1 s and the second 1 s, etc. We are developing a methodology to achieve this, which exploits the particular characteristics of a series of spherical harmonics to represent the data in a way that separates the orientational aspects from the Q-dependent features using an approach already established for multiple component systems [27]. We anticipate that this will enable us to visualize and quantify the information clearly from the time-resolved data. In the measurements reported here, we see no evidence for the dramatic lamellar thickening observed by Yang et al. [28] in some of the closely related crystallization experiments performed under flow and pressure, albeit under isothermal conditions.

The 2d SAXS pattern I(Q,α) is a convolution of the scattering for a perfectly aligned sample I^0^(Q,α) and the orientation distribution of the lamellar crystal orientation D(α) [27]. The Legendre addition theorem allows us to write this convolution as the product of the amplitudes of a series of spherical harmonics for each of these three functions [27,29]: (2)$$I_{2n}(|\underline{Q}|)=\left\{\frac{2\pi }{\left(4n+1\right)}\right\}D_{2n}I_{2n}^{0}(|\underline{Q}|),\alpha )$$
where I_2n_(|Q|), I^0^_2n_(|Q|), and D_2n_ are the spherical harmonic series representing the experimental scattering I(Q,α), the scattering for a perfectly aligned system, and the orientation distribution of the normal to the chain folded lamellar crystals. Only the even order coefficients are required as a consequence of the presence of an inversion centre in the scattering pattern for a weakly absorbing sample [27]. The coefficients for the scattering of a perfectly aligned system have been derived by Lovell and Mitchell [29]. We can write the global orientation parameters <P_2n_> as [27]:(3)$${\langle P_{2n}\rangle }_{Q}=\frac{1}{\left(4n+1\right)P_{2}^{\;m}}\overset{\pi /2}{\underset{0}{\int }}\frac{I\left(\left|\underline{Q}\right|,\alpha \right)\sin\;\alpha P_{2}\;(\cos\alpha )\;d\alpha }{I\left(\left|\underline{Q}\right|,\alpha \right)\sin\alpha \;d\alpha }$$

The orientation parameter <P_2_> is the first component of an even series which describes the orientation distribution function of the normal to the lamellar crystals. If <P_2_> = 0, the distribution is isotropic, and if <P_2_> = 1, the crystals are arranged with the same perfect preferred orientation [29].

Figure 8 shows the azimuthal profile extracted at a fixed value of |Q| for the sixth SAXS pattern shown in Figure 6d. We have used this and equivalent profiles for other SAXS patterns to evaluate the global preferred orientation of the lamellar crystals, or more specifically, the orientation of the surface normal to the chain-folded lamellar crystals.

The objective of this work is to be able to plot the level of preferred orientation of the crystals which have formed in each 1 s time period after the injection has commenced. This will allow us to separate out the two processes taking place in the sequences shown in Figure 6. One is the increasing fraction of formed crystals, and the other is the changing level of the preferred orientation of those crystals. This will allow us to extract the most information from the availability of the in situ time-resolved SAXS data, which will help unravel some of the complexity of the morphology formation in injection moulding. Table 2 shows that using different temperatures of injection leads to different long periods when crystals first form, but which converge to similar values as the crystallinity develops and different levels of orientation are preferred. We might think that these differences arise from the changing cooling times of the melt entering the mould and allowing the extended chains to relax before the crystallization temperature is reached. In interpreting these results, we should bear in mind that, almost certainly, the injected part does not exhibit a homogenous morphology. The time-resolving nature of these experiments allows us to evaluate the polymer which crystallises first and the sequences of the SAXS patterns recorded for the injection temperatures of 205 °C and 210 °C, which show a reduced level of orientation with time. Future experiments will allow us to obtain data at more injection temperatures between 190 °C and 250 °C. The previous work [2] has proposed that initially a thin layer of highly anisotropic polymer crystallises on the mould walls, and then the remaining liquid polymer cools and crystallises in a different manner. Typically, such studies proceed through the post-moulding techniques using sections. We are exploring how we can exploit the availability of time-resolved data to yield the same type of data. This work is in progress, and we hope to report on this in due course.

This work has focused on the development of an industrially relevant injection moulding for use with small-angle X-ray scattering. This focus has allowed us to develop a system with strong relevance to the industry. It would be much more challenging to include a wide-angle X-ray scattering capability, but we will return to this challenge in future work. We note here that although the SAXS-WAXS technique is very powerful, especially for isotropic systems, it is the case that the SAXS experiments are largely performed without compromise, whereas the WAXS experiments generally involve a compromise or restriction in the geometry. This can be a limitation when dealing with anisotropic systems, as in this work. 

## 4. Conclusions

This project has focused on designing, fabricating, and validating an injection moulding system for plastics that will fit on and operate with the ALBA NCD-SWEET SAXS Beamline, which conforms to the current industrial mould design practice. The use of mould inserts with localized thin windows leads to an attenuation of the primary X-ray beam, with 12.4 keV photons of less than 30%. We have shown that we are able to use this system to mould samples of isotactic polypropylene at a range of injection temperatures, which gives rise to a reduced level of preferred orientation in the chain folded lamellar crystals with increasing injection temperatures. By performing these experiments in situ, we were able to separate out the different stages of the crystallization in the mould and their variation with injection temperatures. The design of the mould cavity and the sampling points allow the morphology to be measured along the fill path of the mould in future work, and we will be able to create a 4D model of the development of morphology under industrially relevant conditions with industrially relevant polymers.

## Figures and Tables

**Figure 1 polymers-14-04745-f001:**
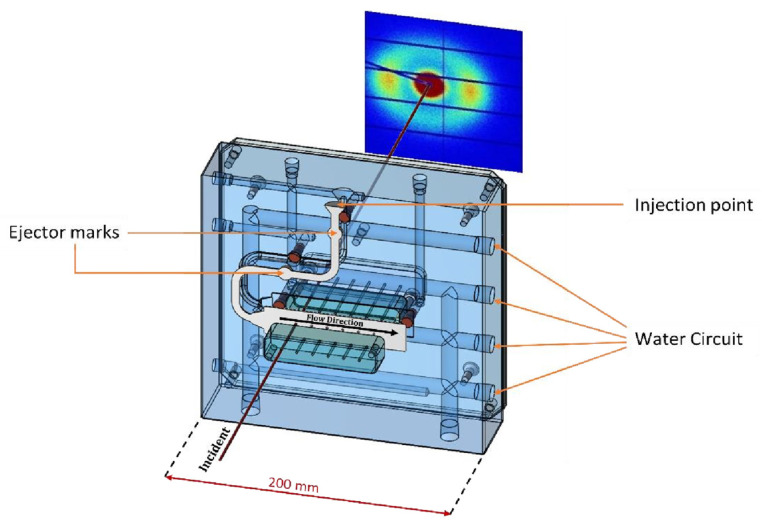
A CAD representation of the components of the mould design used in this work and its relationship with the X-ray beam and detector. The front half of the mould has been removed for clarity.

**Figure 2 polymers-14-04745-f002:**
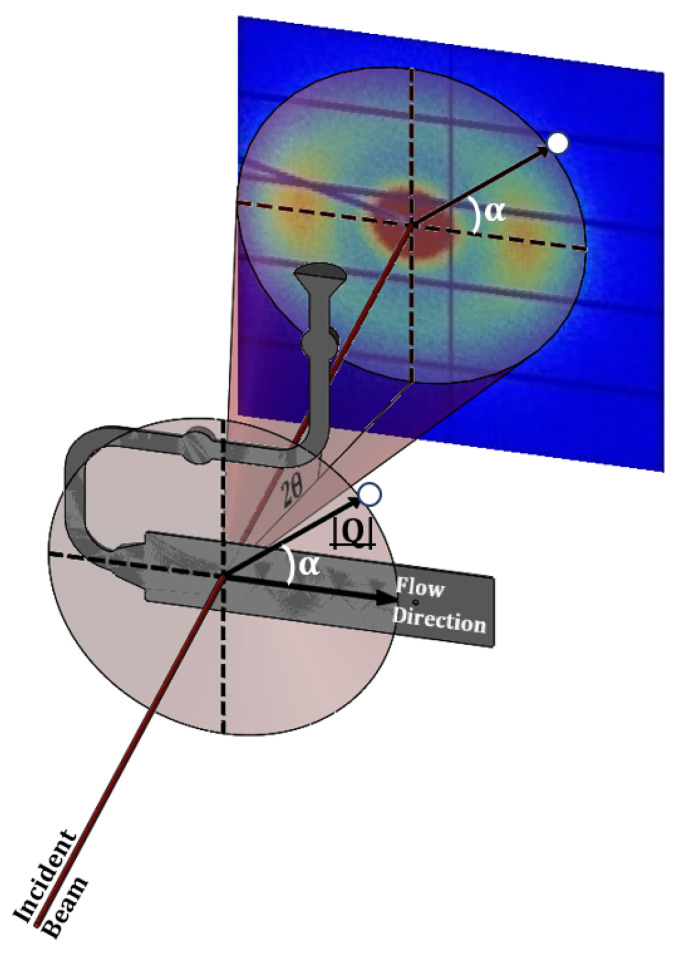
A schematic of the scattering geometry in the Injection Moulding System.

**Figure 3 polymers-14-04745-f003:**
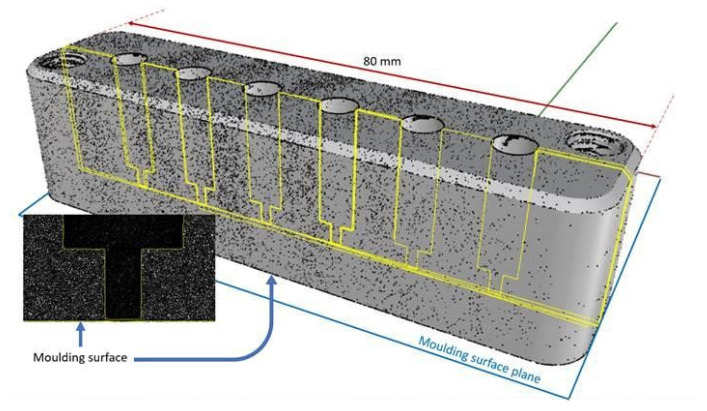
An X-ray CT of one of the mould inserts used in this work. The insert shows an enlarged 139 image showing the thin “window” at the base of each blind hole.

**Figure 4 polymers-14-04745-f004:**
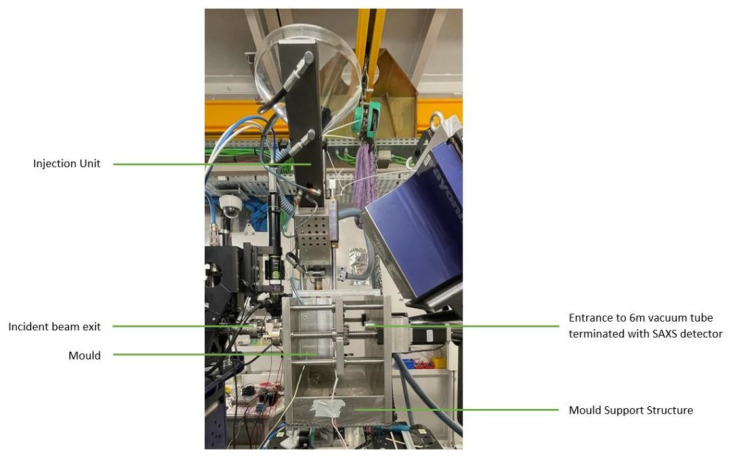
The injection moulding system developed in this work mounted on the NCD-SWEET Beamline. The incident beam enters from the left and the small-angle X-ray scattering detector is mounted 6.7 m to the right and is connected via a vacuum tube which has a conical front piece, which is located very close to the exit side of the mould. The dark blue box which can be observed just above the vacuum tube is the wide-angle X-ray scattering detector which was not used in these experiments.

**Figure 5 polymers-14-04745-f005:**
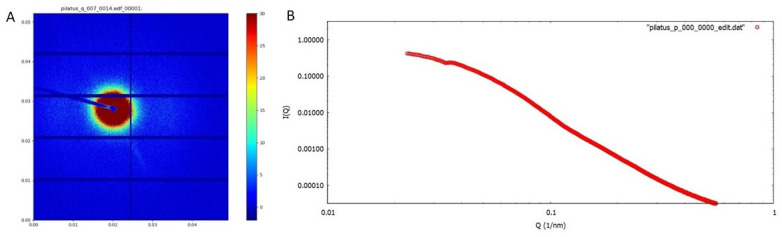
(**A**) The SAXS pattern for the empty mould cavity. The SAXS detector over the range in |Q| from 0.002 to 1.25 Å^−1^ (**B**) The azimuthally averaged data shown in Figure 5A.

**Figure 6 polymers-14-04745-f006:**
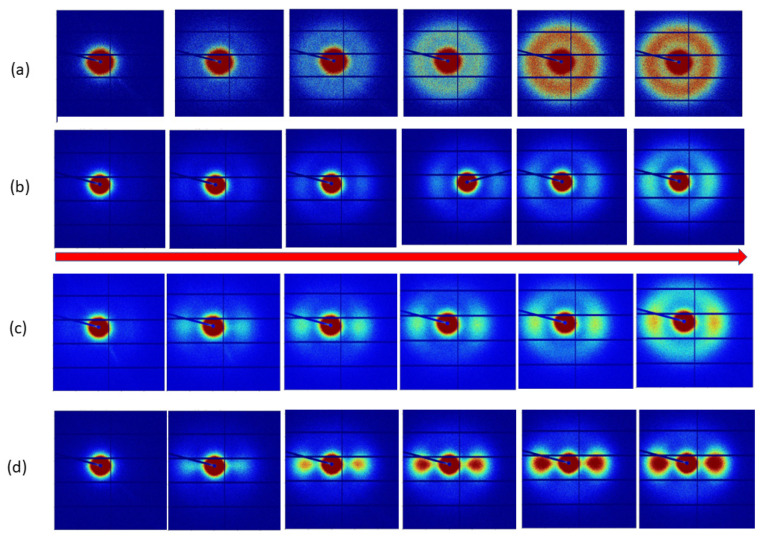
SAXS patterns obtained at approximately 1 s intervals at the first viewing port for injections at (**a**) 250 °C, (**b**) 210 °C, (**c**) 205 °C, and (**d**) 190 °C. The flow in the mould cavity is horizontal as indicated by the horizontal arrow. Time increases from left to right. The intensity mapping is constant within each sequence. The horizontal and vertical dark lines correspond to insensitive areas in the detector as it is made up of rectangular elements. The dark spot in the centre of the pattern corresponds to the area of the detector blocked by the beam stop, and the angled black line corresponds to the mount for the beam stop. The SAXS detector cover the |Q| range from 0.002 Å^−1^ to 0.125 Å^−1^. The data shown here are the raw experimental data; no background has been subtracted so that the quality of the data may be easily judged.

**Figure 7 polymers-14-04745-f007:**
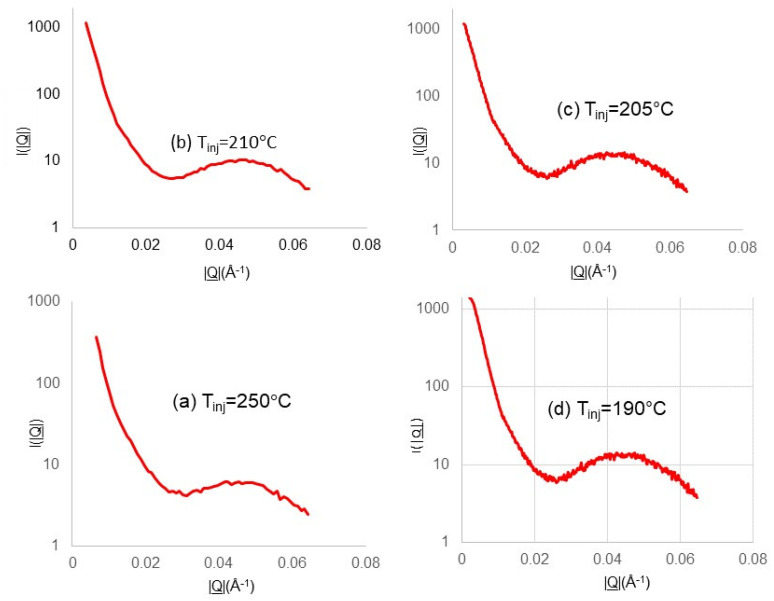
Plots of the section through the 2D SAXS pattern recorded 100 s after the first crystallization for each of the experiments shown in Figure 6. The section was taken in the horizontal direction passing through the centre of the pattern. The data were averaged over a small |Q| interval (5 × 10^−3^Å^−1^) in the vertical axis at each |Q| value in the horizontal direction. The vertical scale is logarithmic. The temperature of injection is shown on each plot.

**Figure 8 polymers-14-04745-f008:**
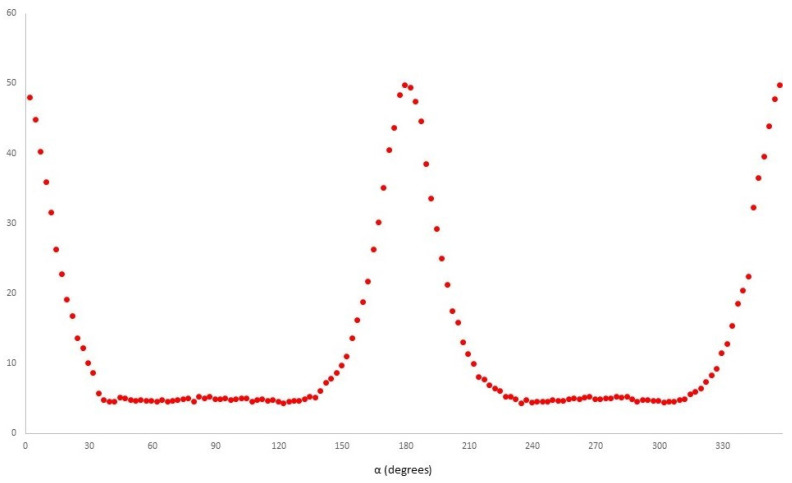
A plot of the azimuthal variation in the intensity in the SAXS patterns at the position of the lamellar stack scattering. Profile derived from the sixth pattern in Figure 6d, with an injection temperature of 190 °C.

**Table 1 polymers-14-04745-t001:** Operational modes of the multi-window mould.

Mode	Number of Data Points per Sampling Point	Time between Pts at Data and Sampling Point (s)	Accumuation Time (s)
A	300	1	1
B	25	18	1
C	Variable	Variable	1

**Table 2 polymers-14-04745-t002:** The Long Period evaluated from SAXS patterns for each injection temperature at times 100 s, 5 s, and 3 s after crystallization commenced and the global orientational parameter <P_2_> and <P_4_> of the lamellar crystals evaluated from SAXS patterns for each injection temperature at a time 100 s after crystallization commenced. The temperature of the mould was fixed at 50 °C.

Injection Temperature °C	Long Period (Å) T = 100 s	Long Period (Å) T = 5 s	Long Period (Å) T = 3 s	<P2> T = 100 s	<P4> T = 100 s
250	140	151	130	0.008	0.002
210	140	144	144	0.043	0.045
205	136	151	158	0.175	0.014
190	141	166	170	0.431	0.231

## Data Availability

The data were obtained using the facilities of the ALBA Synchrotron Light Source and are subject to the generic data management policy at ALBA CELLS, as can be accessed at Microsoft Word-Data_policy_Alba_v1.2_2017.doc (cells.es). The experimental data identifiers are available from the corresponding author after the end of the embargo period.

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
