# Peer review of "Evaluating the Injection Moulding of Plastic Parts Using In Situ Time-Resolved Small-Angle X-ray Scattering Techniques"

_polymers, 2022, doi:10.3390/polym14214745_

Round 1

Reviewer 1 Report

The authors report in this study a development and commissioning of a novel injection moulding setup for studying polymer crystallisation in situ with the help of SAXS analysis. I believe the work is of good quality overall and of interest for a wider community of researchers working in the fields of polymer sciences and applied X-ray analysis. It is sometimes difficult to follow the text due to English grammar. I would recommend for the publication after the following points are addressed:

1) line 96: it would be useful to show directions of vector Q and flow axis in Figure 1 with respect to the injection moulding stage.

2) line 105: space is missing.

3) line 120: should be Figure 1 instead of Figure 2.

4) Photo of the actual moulding staged with the mounted windows is missing. It would help to follow the text in lines 125-133.

5) Figure 2 is not particularly great. Top of the figure with WAXS detector of no use. The injection moulding setup is also not visible. I would suggest to replace it or at least to crop and show the injection moulding stage off-line with arrows showing different components and inlets on the separate photo.

6) line 153: Sentence ‘This was as expected.’ has to be change as it does not make sense.

7) line 169: ‘Each of the 6 windows allows the development of the morphology to be evaluated at different points in the mould cavity.’ Again you need to show a position of all 6 windows on the injection moulding system otherwise it is not clear.

8) line 178: It would be nice to add what difference one would expect to see at different windows (cavities) for non specialist in the field? In other words what advantage this can provide. Since to me the injected polymer in each cavity would behave very similar.

9) Table 1: Could you comment why minimum time in mode B is 18 s. I assume it is limited by the speed of the motor to move from one cavity to another? This time is too large compared to 5s reported in Fig.4 for the crystallisation to be completed.

10) line 193 and Figure 1: Could you please indicate on the Figure 1 where the injection point is located.

11) line 200: Could you estimate the Long period and compare it with the values obtained for this iPP crystallised from the melt (no stress).

12) line 201: it would be good to support this statement by including a 1d plot showing intensity distribution for the peak corresponding to the long period vs azimuthal angle (0-360 deg).

13) line 222: could you please estimate the long period in the oriented iPP crystals and compare this value with the values obtained for the isotropic iPP injected at 250 degC (Fig.4a).

14) line 230: Authors have to analyse 2D SAXS patterns in the sequence in
Fig.4d to calculate the long period to see how it is changing as a function of time and to compare their results with the literature.

15) The authors seem to jump from Fig.4a to Fig.4d in the results section. It would be good to add description of what happened at the temperatures 210 and 205 degC and compare the results with the literature.

15) line 232: I do not see a point of mentioning model developed in ref [20] without comparing it with your results. It would be good to add a sentence on how results from this work compare with ref [20].

16) line 239: I am slighly confused. I understood from the Figure 4 legend that showed images are recorded at each 1s during the injection moulding. And here authors say that this is not the case. Could you please clarify this and change the text accordingly.

17) line 266: agree that there is an interplay between two processes. Authors should comment how this interplay depends on the injection moulding temperatures used in this work and probably try in qualitatevily manner to show the presence of these two processes in oriented patterns in Fig 4. Since it looks like for example that in sequence in Fig. 4c the polymer becomes ‘less’ oriented after 6s compared to 3s i.e. isotropic component appears after 6s.

18) lines 131 and 276: authors should comment on the choice of the photon energy 12.4keV as attentuation length in the material decreases when the photon energy decreases.

19) line 282: this is unclear how the moulding process can be followed along the fill path as window allows to take measurements at one single point at the bottom of the cavity as far as i understood.

20) line 51: authors have to add more references on recent synchrotron-based setups to study polymer crystallisation in situ such as

https://www.mdpi.com/2079-4991/11/10/2652 and https://www.mdpi.com/2073-4360/13/21/3730

21) Is the new setup compatible with the simultanious SAXS and WAXS studies?

Reviewer 2 Report

Major issues

This study presents a new setup developed for in situ SAXS investigations of injection moulded polymers. Besides the description of the setup, the paper contains too few data, basically only the x-ray scattering curve of the empty container, and selected color images of the illuminated detector. Even though the images are meaningful and the message is understandable, this is not enough even for a paper that describes an experimental installation.

Treated data of the polymers along different orientations should be presented in I vs Q coordinates, and a simple data modeling applied to describe the observed features of the scattering curves.

Equations 1,2 are useless, and are not suitable in the actual version of the paper, if they are not used in the further analysis of the data presented in this same paper.

Minor corrections

1. Grammar errors should be corrected in several places, such as lines 95,110, and others.

2. Reference formatting should be unified.

3. The angstrom symbols in the second page should be typed with the appropriate characters of MS Word.

4. The notations of q or Q should be unified.

Round 2

Reviewer 1 Report

Dear Authors,

I am delighted to see that you have addressed all my suggestions which I believed helped to improved the manuscript. Thank you for your efforts here. I am happy to recommend it for publication now. Only please check the quality of Figure 7.

Kind regards

Author Response

See attaached file

Reviewer 2 Report

In the revised version, authors added some more interesting material. The paper can probably be published.  Text and layout formatting is needed according to the mdpi style, and Figure 7 must be redrawn eliminating the many small mistakes, and inserted in a high quality image mode. Angstrom symbols should be changed to proper ones.
